# Barriers for health care access affects maternal continuum of care utilization in Ethiopia; spatial analysis and generalized estimating equation

**Tesfa Sewunet Alamneh**[1]*, **Achamyeleh Birhanu Teshale**[1], **Yigizie Yeshaw**[1,2], **Adugnaw Zeleke Alem**[1], **Hiwotie Getaneh Ayalew**[3], **Alemneh Mekuriaw Liyew**[1], **Zemenu Tadesse Tessema**[1], **Getayeneh Antehunegn Tesema**[1], **Misganaw Gebrie Worku**[4]

**1** Department of Epidemiology and Biostatistics, Institute of Public Health, College of Medicine and Health Sciences, University of Gondar, Gondar, Ethiopia, **2** Department of Physiology, School of Medicine, College of Medicine and Health Sciences, University of Gondar, Gondar, Ethiopia, **3** Department of midwifery, school of nursing and midwifery, college of medicine and health sciences, Wollo University, Dessie, Ethiopia, **4** Department of Human Anatomy, University of Gondar, College of Medicine and Health Science, School of Medicine, Gondar, Ethiopia

* tesfasewunet23@gmail.com

**Data Availability Statement:** Data cannot be shared publicly because of we use third parties DHS data set. Data are available from the measure

## Abstract

### Background

Although Ethiopia had made a significant change in maternal morbidity and mortality over the past decades, it remains a major public health concern. World Health Organization designed maternal continuum of care to reduce maternal morbidity and mortality. However, majority of the mothers didn't utilize the maternal continuum of care. Therefore, this study aimed to assess the spatial distribution of incomplete utilization of maternal continuum of care and its associated factors in Ethiopia.

### Methods

This study was based on 2016 Demographic and Health Survey data of Ethiopia. A total weighted sample of 4,772 reproductive aged women were included. The study used ArcGIS and SaTScan software to explore the spatial distribution of incomplete utilization of maternal continuum of care. Besides, multivariable Generalized Estimating Equation was fitted to identify the associated factors of incomplete utilization of maternal continuum of care using STATA software. Model comparison was made based on Quasi Information Criteria. An adjusted odds ratio with 95% confidence interval of the selected model was reported to identify significantly associated factors of incomplete utilization of maternal continuum of care.

### Results

The spatial analysis revealed that incomplete utilization of maternal continuum of care had significant spatial variation across the country. Primary clusters were detected at Somali, North-Eastern part of Oromia, and East part of Southern Nation Nationalities while

DHS website/ www.dhsprogram.com for researchers who meet the criteria for access to confidential data. Furthermore, the researchers can access the birth recode file of EDHS dataset from https://www.dhsprogram.com/data/dataset_admin/login_main.cfm after getting written consent from DHS website. Variables extracted for this analysis were v001 (Cluster variable), v005 (used for generating weighting variable; v0005/1000000), ANC visit, institutional delivery, Post Natal care visit, age of the women, residence, marital status, religion, maternal education, wealth index, currently working, mass media exposure, number of births/parity, contraceptive use, barriers for accessing health care (women reported at least one challenge of health care access (money, distance, companionship, and permission), and wanted pregnancy.

**Funding:** The author(s) received no specific funding for this work.

**Competing interests:** The authors have declared that no competing interests exist.

**Abbreviations:** ANC, Antenatal care; AOR, Adjusted Odds Ratio; CI, Confidence Interval; COR, Crude Odds Ratio; DHS, Demographic and Health Survey; EA, Enumeration Areas; EDHS, Ethiopia Demographic and Health Survey; GEE, Generalized Estimating Equation; ICC, Intra-class Correlation Coefficient; LLR, Log Likelihood Ratio; PNC, Post Natal Care; QIC, Quasi Information Criteria; SNNPR, Southern Nations, Nationalities, and People's Region; WHO, World Health Organization.

secondary clusters were detected in the Central Amhara region. In multivariate GEE, rural residency, secondary education, higher education, Protestant religious follower's, Muslim religious follower's, poorer wealth index, richer wealth index, richest wealth index, currently working, having barriers for accessing health care, and exposure for mass media were significantly associated with incomplete utilization maternal continuum of care.

## Conclusion

Incomplete utilization of maternal continuum of care had significant spatial variations in Ethiopia. Residence, wealth index, education, religion, and barriers for health care access, mass media exposure, and currently working were significantly associated with incomplete utilization of maternal continuum of care. Therefore, public health interventions targeted to enhance maternal service utilization and women empowerment in hotspot areas of incomplete utilization of maternal continuum of care are crucial for reducing maternal morbidity and mortality.

## Background

Maternal mortality has long been a major public health concern, which is defined as the death of a woman while pregnant or within 42 days of after termination of pregnancy [1]. Globally, an estimated 303,000 women dying each year as a result of pregnancy and child birth-related complications [2]. Of those, 99% were accounted by developing countries including Ethiopia [3]. Although maternal deaths worldwide declined by 44% between 1990 and 2015, the Millennium Development Goal 5a, which was aimed at 75% reduction in maternal mortality, was not achieved [3, 4]. In late 2015, Sustainable Development Goal (SDG) 3 were launched to address universal health coverage or ensure healthy lives and promoting well-being for all at all ages. Reduction of the global Maternal Mortality Rate (MMR) and ending of preventable deaths of newborns and under 5 years of age children are the particular focus in SDG 3 [5]. Thus, maternal continuum of care is a key tool to achieve the SDG 3 global targets [6, 7]. Despite Ethiopia had made significant change in reducing maternal and child morbidity and mortality over the past decades, maternal and child health problems had remained a major public health problem in the country; 412 maternal deaths per 100,000 live births, 57% had four and above Ante Natal Care (ANC) visit, 53% women hand delivered at home, and 66% women had no postnatal care service.

The finding of previous studies in the globe revealed several factors that are associated with utilization of maternal continuum of care. These include: women age, place of residency, marital status, current working status, religion, wealth index, mass media exposure, distance from the health facility, being informed about pregnancy complications, number of children (parity), and age at first birth [8–17]

Incomplete utilization of maternal continuum of care had also varied across the country. It is high among rural and poor societies. Thus, identification of the areas having incomplete utilization of maternal continuum of care using geographical information system (GIS) and spatial scan statistical analysis (SaTScan) has become an essential to guide focused public health interventions. Besides, previous studies in Ethiopia have been focused on the magnitudes and determinants of utilization of maternal continuum of care using ordinary logistic regression models despite the hierarchical structure of the Ethiopia Demographic and Health Survey

(EDHS) data. Thus, this data had nested effect and violate independent and equal variance assumptions that result in biased estimates. The findings of previous studies were insufficient and limited to capture the spatial distribution of incomplete utilization of maternal continuum of care. Therefore, this study aimed to investigate the spatial distribution and associated factors of incomplete utilization of maternal continuum of care among reproductive aged women in Ethiopia using spatial analysis and Generalized Estimating Equations. Conducting this study will help to make decision on maternal health service utilization based on the available evidences. Moreover, the result of this study could support policy makers, clinicians, and programmers to design intervention for achieving the targeted SDG 3.

## Methods and materials

### Study design, setting and period

The present study used 2016 EDHS data. The survey was collected every 5 years to assess population and health indicators at the national and regional levels using a structured, validated, and standardized questionnaire. It was also conducted for four times in Ethiopia. Hence, the 2016 EDHS is the latest and the fourth survey conducted in the country. Ethiopia is an East African country with an estimated population of 109.2 million that makes second most populous country in Africa [18]. Ethiopia is federally decentralized in to nine regions and two city administrations and the regions are further divided into zones, and zones into administrative units called districts [19]. The district again subdivided into kebele which is the lowest administrative unit. Regarding to the health care system in Ethiopia, the fourth health sector development plan introduced a three-tier health-service delivery system. This system were arranged by including Primary health care unities (i.e. health posts and health centers) and primary hospitals at primary level, general hospitals at secondary level, and specialized hospitals at tertiary level [20].

### Study population and sampling technique

All reproductive aged women who were booked for ANC service and giving birth within 5 years preceding the 2016 survey in Ethiopia were the source population, while all reproductive aged women who were booked for ANC service and giving birth in the selected Enumeration Areas (EAs) within 5 years before the 2016 survey were the study population. The most recent birth characteristics was used for those who give more than one birth within five years preceding the survey. A two stage stratified cluster sampling technique were employed to select study participants. Stratification of regions into urban and rural areas were considered. In the first stage, 645 EAs (202 from urban area) were selected using probability sampling proportional to the EAs size and with independent selection in each sampling stratum. In the second stage, 28 households from each cluster were selected with an equal probability of selection from the household listing [21]. A total of 4,772 weighted reproductive aged women were included in the study.

### Measurement of variables

The response variable for this study was maternal continuum of care. Maternal continuum of care is a series of cares provided for mothers during the three periods of maternity [11, 17]. It is a composite variable obtained from ANC, institutional delivery, and Post Natal care (PNC) services. The response variable for the $i^{th}$ mother from $j^{th}$ cluster (EAs) was represented by a random variable Yij, with two possible values coded as 1 and 0. The outcome variable of the ith mother in the jth cluster (Yij) = 1 if ith mother had incomplete maternal continuum of care

or if the women had not utilize one of the three maternity services (i.e. 4 and above ANC visits, institutional delivery or postnatal checkup) and Yij = 0 if the mother had complete continuum of maternal care (if the women's had utilize all the three maternity services).

Age of the women, residence, marital status, religion, maternal education, wealth index, currently working, mass media exposure, number of births/parity, contraceptive use, barriers for accessing health care (women reported at least one challenge of health care access (money, distance, companionship, and permission) considered as having barriers of for accessing health care while if a woman didn't report none of the above challenges were considered as no barriers for accessing health care) [22], wanted pregnancy were included as independent variables.

## Data management and statistical analysis

After accessing the data, the variables of the study were extracted from birth recorded data set of EDHS data, data cleaning, and recoding were conducted in STATA version 14.1. The data were weighted using sampling weight and complex survey design was used to adjust for unequal probability of selection due to the sampling design employed in EDHS data.

## Spatial analysis

The spatial analysis was done using ArcGIS V.10.7 and SaTScan V.9.6 software. These study conducts the spatial autocorrelation, hot spot analysis, spatial interpolation, and SaTScan analysis of incomplete utilization of maternal continuum of care.

## Spatial autocorrelation analysis

Spatial autocorrelation was conducted to test whether the spatial distribution of incomplete utilization of maternal continuum of care was randomly distributed or not. The Global Moran's I statistics which ranges from −1 to +1 was used to measure whether the distribution of incomplete utilization of maternal continuum of care was dispersed, random, or clustered in the study area [23]. The statistic values close to −1 indicate spatial distribution of incomplete utilization of maternal continuum of care is dispersed, a statistic close to value 0 indicates incomplete utilization of maternal continuum of care is randomly distributed, and a statistic close to +1 means the spatial distribution of incomplete utilization of maternal continuum of care was clustered [24].

## Hotspot analysis

Getis-Ord Gi* statistics was used to identify areas with higher rates of incomplete utilization of maternal continuum of care (significant hot spots areas), and areas with lower rates of incomplete utilization of maternal continuum of care (cold spot areas) [25].

To assess the presence of statistically significant spatial clusters of incomplete utilization of maternal continuum of care, Bernoulli based spatial scan statistical analysis with circular window was done. Women with incomplete utilization of maternal continuum of care were taken as cases and women with complete utilization of maternal continuum of care was taken as controls to fit the Bernoulli based model. The default maximum spatial cluster size of less than 50% of the population was used as an upper limit for the identification of both small and large clusters. Log Likelihood Ratio (LLR) test was used to the significance of the clusters and the 999 Monte Carlo replications were used to calculate p values and to rank using their LLR test. Finally, the primary cluster was chosen as the spatial window when it has greatest LLR test [26].

## Spatial interpolation

To predict incomplete utilization maternal continuum of care in unsampled areas in the country based on the data in sampled clusters /EA, spatial interpolation technique was employed. Although various spatial interpolation techniques are available, this study used an Empirical Bayesian Kriging (EBK) technique which are considered the best methods since it incorporates spatial autocorrelation and statistically optimize the weight [27].

## Associated factors of incomplete utilization of maternal continuum of cares

In the EDHS data, women are nested within a cluster/EAs and those who reside within the same clusters have similar characteristics compared to those from another clusters. This violates the independence and equal variance assumptions of the ordinary logistic regression model. Thus, Intra-class Correlation Coefficient (ICC) was computed to measure the variability between clusters after fitting a model without any covariate. It quantifies the degree of heterogeneity of incomplete utilization of maternal continuum care between clusters (the proportion of variance explained by the between cluster variability). It also computed as;

$ICC = \frac{\sigma_\mu^2}{\sigma_\mu^2 + \pi^2/3}$; Where: the ordinary logit distribution has variance of $\pi^2/3$, $\sigma_\mu^2$ indicates the cluster variance [28]. The calculated ICC was 36.37%, showed that about 36.37% of the variation in incomplete continuum of care was explained by the between cluster variation. This implies the need to take into account between-cluster variability by using advanced modelling techniques. Therefore, Generalized Estimating Equation (GEE) model was fitted to identify the associated factors of incomplete utilization of maternal continuum of care among reproductive aged women [29]. It is a marginal model that considers working correlation structure among clusters that estimates a robust standard error and also controlled for within-cluster correlations. Generalized Estimating Equation (GEE) model was fitted with a logit link function and binomial family with independent and exchangeable working correlation structures. Quasi Information Criteria (QIC) was used to select the best-fitted model. The model with exchangeable correlation structure was selected as the best fitted model since it had smaller QIC value. Variables with p-value <0.2 in the bi-variable GEE were considered for the multivariable GEE model. To assess the strength of association between outcome variable and independent factors both crude and adjusted odds ratio with a 95% Confidence Interval (CI) were computed. Variables having less than 5% p-value in the multivariable GEE model were considered as the associated factors with the incomplete utilization of maternal continuum care.

## Ethics consideration

This study is a secondary data analysis from the EDHS data, so it does not require ethical approval. For conducting this study, online registration and request for measure DHS were conducted. The dataset was downloaded from DHS on-line archive after getting approval to access the data. Regarding the shape file, online request and registration was done with the following adress https://africaopendata.org/dataset/ethiopia-shapefiles. All methods were carried out in accordance with the Declaration of Helsinki.

## Results

A total of 4,772 study participants were included in these study. Of them, 252 (5.27%) were teenagers. More than three in four, 3,897 (81.66%) were rural dweller's. Moreover, 2,581

(54.07%) mother was not attending formal education. Regarding accesses to health care services, nearly one in third, 3,336 (69.90%) had barriers for accessing health cares (Table 1).

## Magnitudes of maternal continuum of cares

Among the study participants, only 247 (5.17%) had complete maternal continuum of care (i.e. had four and above ANC visit, institutional delivery, and PNC services) (Table 2).

## Spatial analysis

**Spatial autocorrelation of incomplete utilization of maternal continuum of care.** The spatial analysis found that the spatial distribution of incomplete utilization of maternal continuum of care was significantly varied across the country (Global Moran's I = 0.46, p<0.001) (Fig 1). The highest prevalence of incomplete utilization of maternal continuum of care was identified in the majority of Tigray, Dire Dawa, Harar, Oromia, and SNNP regions whereas the lowest prevalence of incomplete maternal continuum of care was detected in Afar and Somali regions (Fig 2).

**Hotspot analysis of incomplete utilization of maternal continuum care.** In hot spot analysis, a significant hot spot clusters of incomplete utilization of maternal continuum of care were identified in South Eastern part of Gambela, Addis Ababa, Central part of Oromia, Dire Dawa, and Harari regions. While significant cold spot areas were in Afar, Eastern part of Amhara, and western part of Oromia region (Fig 3).

A total of 210 significant clusters of incomplete utilization of maternal continuum of care were detected in spatial scan statistical analysis, of them 186 clusters were primary clusters (most likely clusters). The primary cluster spatial window was founded in the Somali, eastern part of Oromia, and south East part of SNNP regions, with geographical location of 4.180558 N and 42.052871 E with a 600.02 km radius, a relative risk of 1.07 and a log likelihood ratio of 39.52 (p<0.001). This revealed that a woman within the spatial window had 1.07 times higher risk of incomplete utilization of maternal continuum care compared to women outside the window (Table 3). The red circular ring contains the primary clusters of incomplete utilization of maternal continuum of care which are the most statistically significant spatial window in LLR test. Moreover, the secondary clusters were identified in the central Amhara region (Fig 4).

**Kriging interpolation of incomplete utilization of maternal continuum of care.** In the EBK interpolation, high prevalence of incomplete utilization of maternal continuum of care were predicted at Dire Dawa, south western part of Tigray, southern part of SNNP, western part of Benishangul-Gumz, central and western part of Oromia Whereas, the predicted lowest prevalence of incomplete utilization of maternal continuum of care were detected in the Somali and Afar (Fig 5).

**Factors associated with incomplete utilization of maternal continuum of care.** In the bi-variable analysis maternal age, ever had terminated pregnancy, and was the pregnancy wanted were not associated with incomplete utilization of maternal continuum of care at a p-value less than 0.2. The multivariable GEE model revealed that variables such as residence, educational level, religion, wealth index, mass media exposure and barriers for health care accesses were significantly associated with incomplete utilization of maternal continuum of care at a 5% level of significance.

This study revealed that the odds off having incomplete utilization of maternal continuum of care were 1.67 (AOR = 167, 95%CI; 1.02, 2.74) times higher for rural residents as compared to their counterparts. Educational level was also significantly associated with incomplete utilization of maternal continuum of care with the likelihood of incomplete utilization of maternal

**Table 1.** Background characteristics of reproductive aged women in Ethiopia, 2016 EDHS.

| Variables | Category | Frequency | Proportion(%) |
|---|---|---|---|
| Age | 15–19 | 252 | 5.27 |
| | 20–34 | 3,481 | 72.95 |
| | 35–49 | 1,040 | 21.78 |
| Region | Tigray | 486 | 10.18 |
| | Afar | 37 | 0.77 |
| | Amhara | 1,104 | 23.13 |
| | Oromia | 1,608 | 33.69 |
| | Somali | 119 | 2.48 |
| | Benishangul | 56 | 1.17 |
| | SNNPR | 1,115 | 23.36 |
| | Gambela | 16 | 0.32 |
| | Harari | 14 | 0.28 |
| | Addis Ababa | 192 | 4.02 |
| | Dire Dawa | 30 | 0.61 |
| Residence | Urban | 876 | 18.34 |
| | Rural | 3,897 | 81.66 |
| Educational status | No formal education | 2,581 | 54.07 |
| | Primary | 1,577 | 33.05 |
| | Secondary | 388 | 8.12 |
| | Higher | 227 | 4.76 |
| Religion | Orthodox | 2,030 | 42.54 |
| | Protestant | 1,050 | 21.99 |
| | Muslin | 1,573 | 32.96 |
| | Other | 120 | 2.50 |
| Wealth index | Poorest | 795 | 16.65 |
| | Poorer | 936 | 19.60 |
| | Middle | 997 | 20.88 |
| | Richer | 967 | 20.26 |
| | Richest | 1,079 | 22.61 |
| Terminated pregnancy | No | 4,332 | 90.78 |
| | Yes | 440 | 9.22 |
| Contraceptive use | Non- users | 2,774 | 58.13 |
| | Traditional method | 27 | 0.56 |
| | Modern method | 1,971 | 41.31 |
| Health insurance | No | 4,531 | 94.94 |
| | Yes | 242 | 5.06 |
| Marital status | Single | 90 | 1.87 |
| | Married | 4,481 | 93.91 |
| | Widowed | 54 | 1.12 |
| | Divorced | 147 | 3.10 |
| Currently working | No | 3,260 | 68.30 |
| | Yes | 1,513 | 31.70 |
| Parity | 1–4 | 3,213 | 67.32 |
| | ≥ 5 | 1,560 | 32.68 |
| Ever had terminated pregnancy | No | 4,332 | 90.78 |
| | Yes | 440 | 9.22 |
| Wanted pregnancy | Then | 3,622 | 75.91 |
| | Later | 826 | 17.31 |
| | No more | 324 | 6.79 |

(*Continued*)

**Table 1.** (Continued)

| Variables | Category | Frequency | Proportion(%) |
|---|---|---|---|
| Mass media exposure | No | 3,570 | 74.82 |
| | Yes | 1,202 | 25.18 |
| Health care access barrier | No barrier | 1,437 | 30.10 |
| | Have barrier | 3,336 | 69.90 |

**Table 2. Frequency distribution of maternal continuum of care among reproductive age women in Ethiopia, 2016 EDHS.**

| Variable | Category | Utilization of maternal continuum care | | |
|---|---|---|---|---|
| | | Complete | Incomplete | p-value |
| Maternal age | <20 | 14 | 236 | 0.570 |
| | 20–34 | 216 | 3,216 | |
| | > = 35 | 73 | 957 | |
| Residence | Urban | 176 | 1,219 | <0.0001 |
| | Rural | 127 | 3,190 | |
| Educational status | No formal education | 84 | 2,256 | <0.0001 |
| | Primary | 101 | 1,442 | |
| | secondary | 63 | 458 | |
| | Higher | 55 | 253 | |
| Religion | Orthodox | 188 | 1,668 | <0.0001 |
| | Protestant | 30 | 825 | |
| | Muslim | 84 | 1,852 | |
| | Other | 1 | 64 | |
| Wealth index | poorest | 20 | 1,048 | <0.0001 |
| | poorer | 33 | 728 | |
| | middle | 27 | 679 | |
| | Richer | 40 | 631 | |
| | Richest | 183 | 1,323 | |
| Contraceptive use | Non-user | 148 | 2,792 | <0.0001 |
| | Traditional user | 10 | 34 | |
| | Modern user | 145 | 1,583 | |
| Health insurance use | No | 276 | 4,220 | 0.001 |
| | Yes | 27 | 189 | |
| Marital status | Single | 15 | 92 | 0.044 |
| | Married | 272 | 4,083 | |
| | Widowed | 4 | 55 | |
| | Divorced | 12 | 179 | |
| Currently working | No | 155 | 2,985 | <0.0001 |
| | Yes | 148 | 1,424 | |
| Ever had terminated pregnancy | No | 271 | 3,990 | 0.550 |
| | Yes | 32 | 419 | |
| Parity | <5 | 247 | 3,047 | <0.0001 |
| | > = 5 | 56 | 1,362 | |
| Was the pregnancy wanted | Then | 244 | 3,521 | 0.838 |
| | Later | 45 | 650 | |
| | No more | 14 | 238 | |
| Barriers for accessing health care | no | 165 | 1,536 | <0.0001 |
| | Yes | 138 | 2,873 | |
| Mass media exposure | No | 136 | 3,174 | <0.0001 |
| | Yes | 167 | 1,235 | |

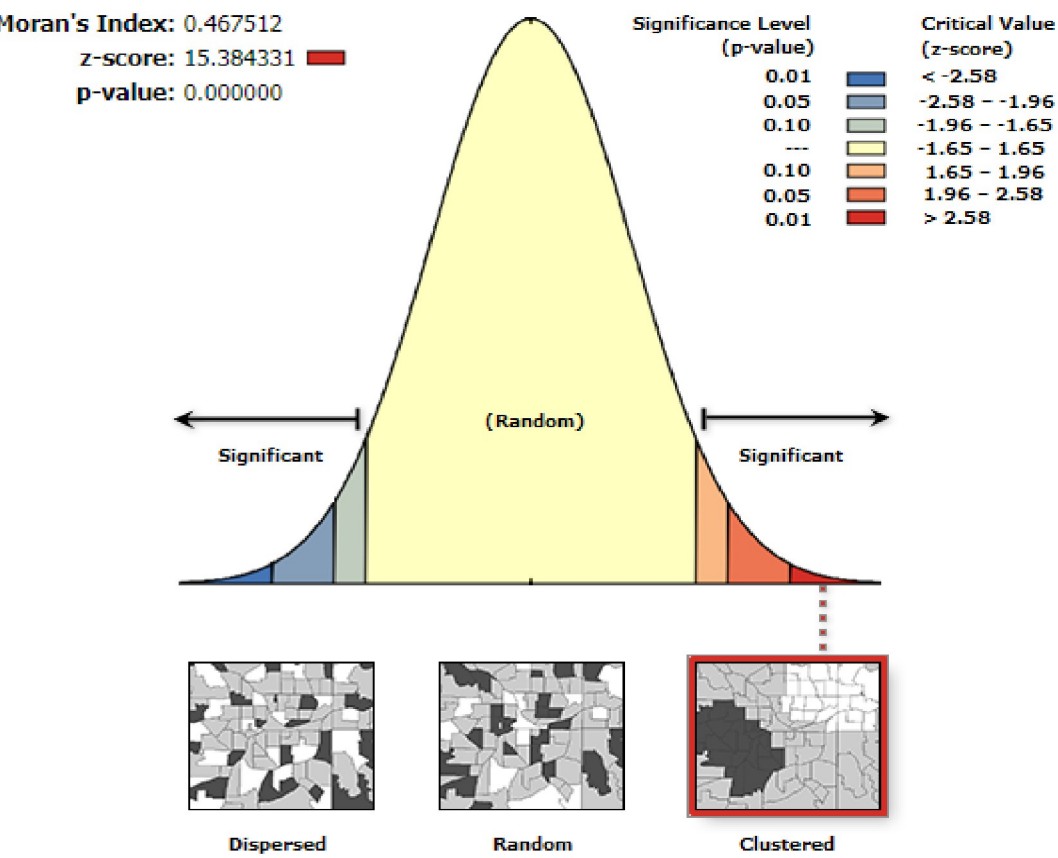

**Fig 1. Global spatial autocorrelation of incomplete utilization of maternal continuum of care in Ethiopia, produced using Arc GIS version 10.7.**

continuum of care were 43% (AOR = 0.57, 95%CI; 0.37, 0.86) and 52% (AOR = 0.48, 95% CI;0.30, 0.77) lower for secondary and higher education respectively as compared with reproductive aged women who have no formal education. Moreover, the odds of having incomplete utilization of maternal continuum of care among protestant and Muslim religious followers were 1.93 (AOR = 1.93, 95%CI;1.25, 2.99) and 1.46 (AOR = 1.46, 95%CI; 1.06, 2.01) times higher as compared to orthodox followers, respectively. In addition, wealth index was significantly associated with incomplete utilization of maternal continuum of care. As compared with poorest reproductive women in their wealth index, the chance of having incomplete utilization of maternal continuum of care was 48% (AOR = 0.52, 95%CI;0.29, 0.95), 61% (AOR = 0.39, 95%CI;0.22, 0.72), and 53% (AOR = 0.47, 95%CI;0.24, 0.92) lower for poorer, richer, and richest respectively. A reproductive aged woman who have barriers for accessing health care had 1.67 (AOR = 1.27, 95%CI;1.12, 1.67) higher odds of having incomplete utilization of maternal continuum of care as compared with women have no barriers. The likelihood of having incomplete utilization of maternal continuum of care among mass media exposed and currently working women were 25% (AOR = 0.75; 0.55, 0.83) and 28% (AOR = 0.72, 95% CI;0.56, 0.93) lower as compared to their counterparts, respectively (Table 4).

## Discussion

Despite maternal and child mortality had declined overtime, till it is a major public health concern in the world particularly in developing countries like Ethiopia [4, 30]. Though WHO

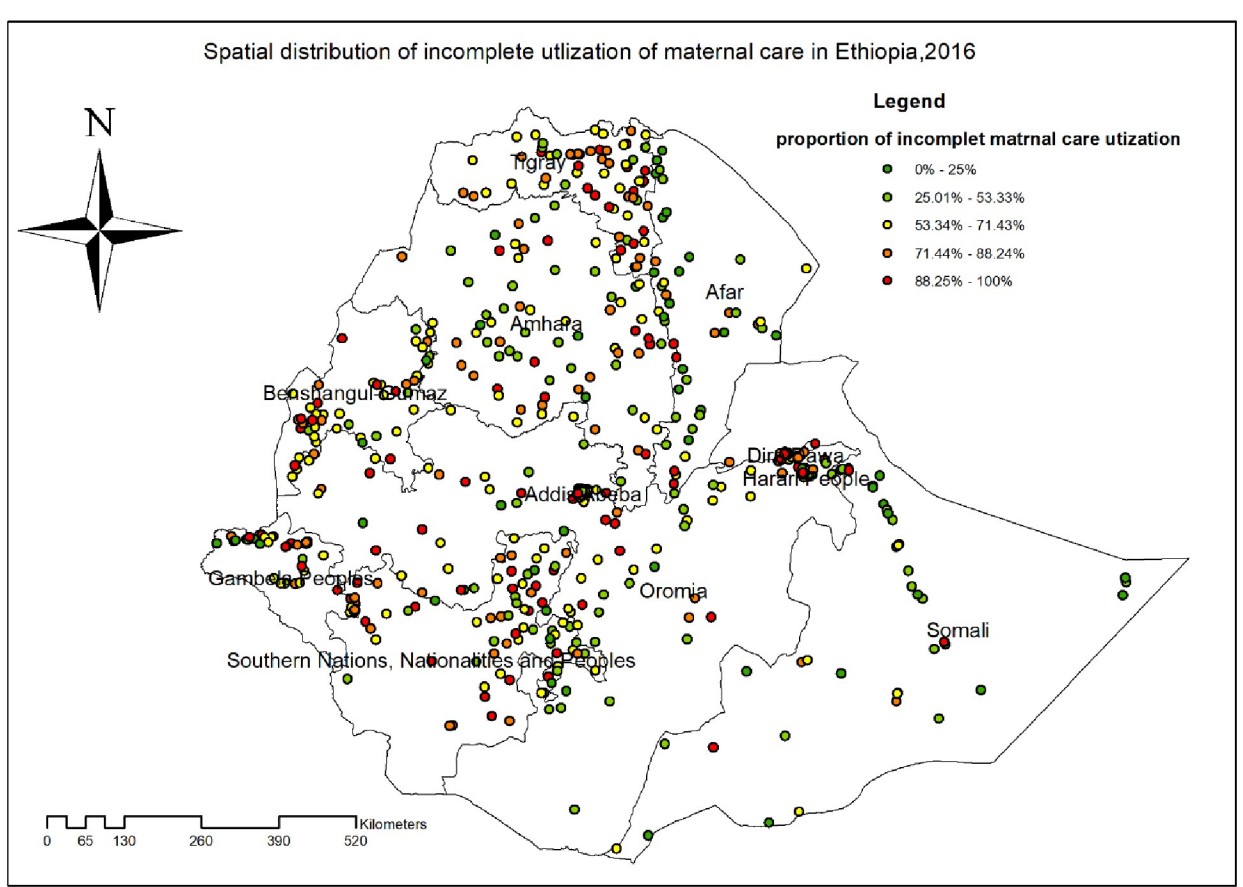

**Fig 2. Spatial distribution of incomplete utilization of maternal continuum of care among reproductive aged women in Ethiopia, produced using Arc GIS version 10.7.**

adopts continuum of Care as a key strategy to tackle maternal and child morbidity and mortality [31, 32]. Thus, this study aimed at investigating the spatial distribution and associated factors of incomplete utilization of continuum of care in Ethiopia.

This study found that the magnitudes of incomplete utilization of maternal continuum of care was 94.83%. Its magnitudes had shown significant variation across the country. Significant hot spot areas of incomplete utilization of maternal continuum of care was detected in Somali, east Oromia, and south East part of southern nation nationalities and people's, and central Amhara regions of Ethiopia. This could be explained by high dropout rate in maternal continuum of care service utilization in the country specially in the detected hot spot areas [11]. In addition, the areas are more rural, had low exposure for mass media, and had a big problem in accessing health care's that magnifies the dropout rate from complete utilization of maternal continuum of care and creates disparity in the availability and accessibility of maternal health services [33].

In generalized estimating equations, incomplete utilization of maternal continuum of care was significantly affected by different factors. It was founded that the odds of having incomplete utilization of maternal continuum of care among women residing in rural area were higher than in urban areas. This finding was supported by previous studies done in Egypt, Nepal and Cambodia [9, 34, 35]. This could be explained by women's in rural areas had relatively poor healthcare-seeking behavior, had limited accessibility, and availability of health

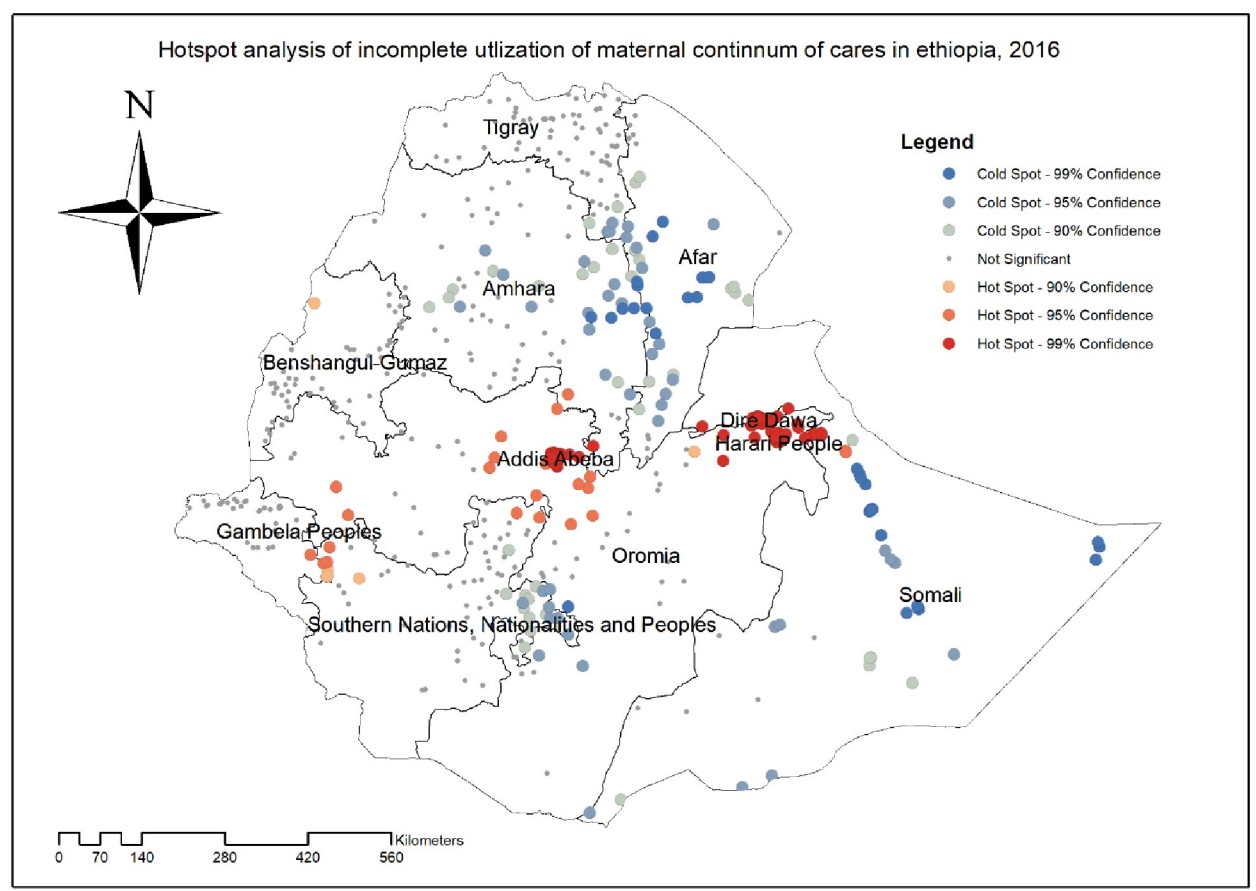

**Fig 3. Hot spot analysis of incomplete utilization of maternal continuum of care using Getis Ord Gi* statistics in Ethiopia, produced using Arc GIS version 10.7.**

facilities [36, 37]. Besides, women residing in rural areas have limited access for education and low chance of getting health information than women residing in urban areas [38, 39].

This study also revealed that odds of having incomplete utilization of maternal continuum of care was lower for women with secondary and higher education as compared to women's not attending formal education. This finding was in consistent with studies done at Egypt and Gahan [34, 40]. This might be related with educated women had better access to health facilities, good health seeking behavior, and better awareness on the maternal health services [41, 42]. Moreover, studies also suggested that illiterate women are economically unstable and they may fail to receive adequate maternal health services during pregnancy, delivery and postpartum periods [43].

**Table 3. SaTScan result analysis, EDHS 2016.**

| Cluster / enumeration area identified | Coordinate/radius | Population | Case | RR | LLR | P value |
|---|---|---|---|---|---|---|
| 1 (186) | 4.18 N, 42.05 E / 600.02 km | 1455 | 1423 | 1.07 | 39.52 | < 0.0001 |
| 2 (24) | 11.29 N, 38.41E/ 115.42 km | 169 | 169 | 1.07 | 11.69 | 0.0044 |
| 3 (52) | 8.45 N, 36.34 E/ 183.04 km | 411 | 402 | 1.05 | 9.12 | 0.050 |
| 4 (8) | 8.20 N, 34.29/ 32.91 km | 61 | 61 | 1.07 | 4.17 | 0.99 |
| 5 (11) | 11.34 N, 35.13 E/ 125.55 km | 61 | 61 | 1.07 | 4.17 | 0.99 |

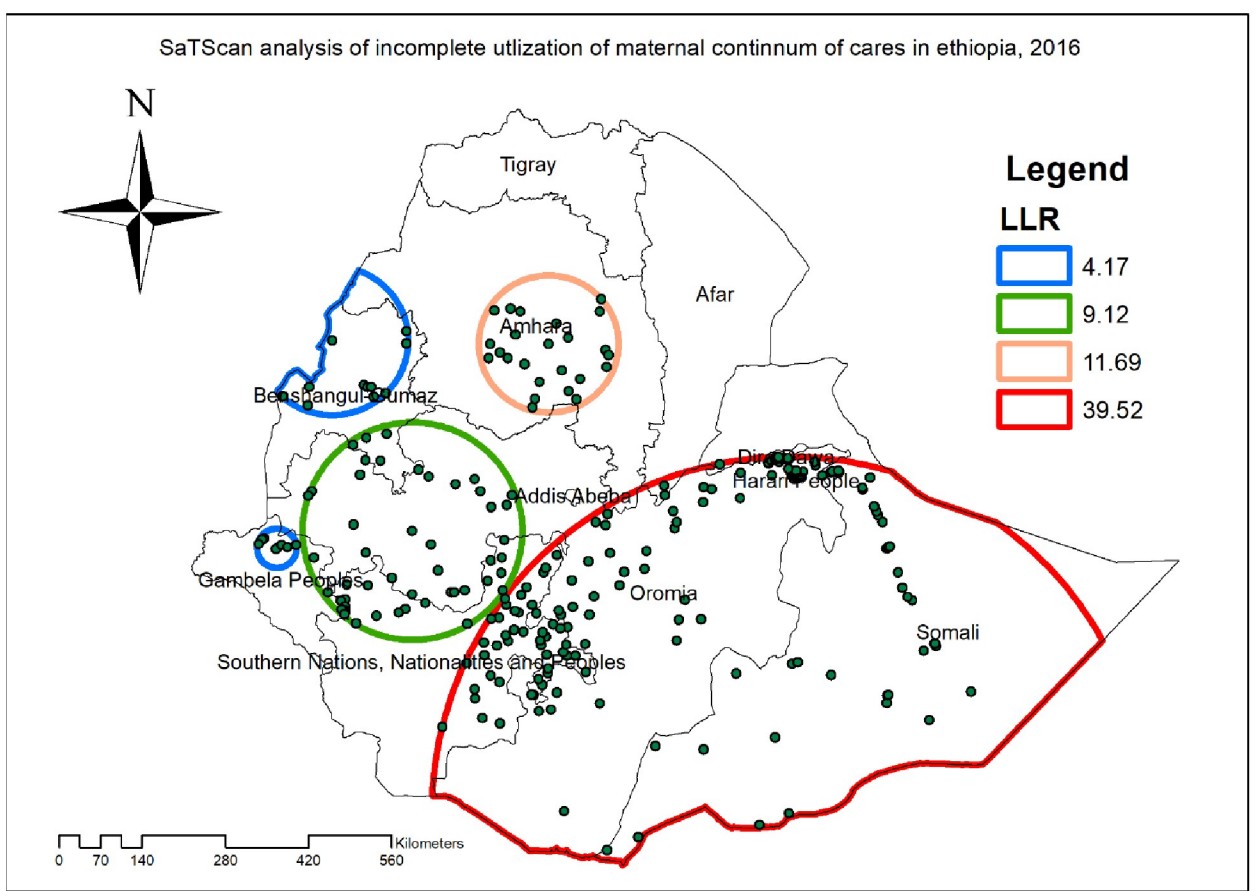

**Fig 4. Kriging interpolation of incomplete utilization of maternal continuum of care in Ethiopia, produced using Arc GIS version 10.7.**

Wealth index were an independent and significant predictor of utilization of maternal continuum of care. The likelihood of having incomplete utilization of maternal continuum of care was lower for richest, richer and poorer as compared with poorest women. This finding was in agreement with studies conducted in Tanzania, Nigeria, and Ethiopia [11, 15, 44]. The possible explanation for the finding could be that women from better wealth index have better socioeconomic status that cerates good access for health facilities and enhance the utilization of health services [45]

In consistent with studies don at Egypt and Nepal [34, 35], exposure for mass media lowers the chance of incomplete utilization maternal continuum of care. This could be linked with mass media is an important means of disseminating information concerning maternal health that may increases knowledge, attitude and practice of women towards maternal health service utilization [35]. Besides, women having mass media exposure have better knowledge on danger signs of pregnancy and child birth that drives for utilizing maternal health services [46].

This study identified that barriers for access to health facility was an important and significant factor for incomplete utilization of maternal continuum of care. The odds of having incomplete utilization of maternal continuum of cares were higher for women having difficulties for accessing health facilities as compared with their counter parts. This finding was supported in studies conducted in Ethiopia and Nigeria [44, 47]. The possible justification for this finding might be barriers for accessing health facility such as difficulties in obtaining money, long distance travel to health facility, not wanting to go alone may result in difficulty to arrive

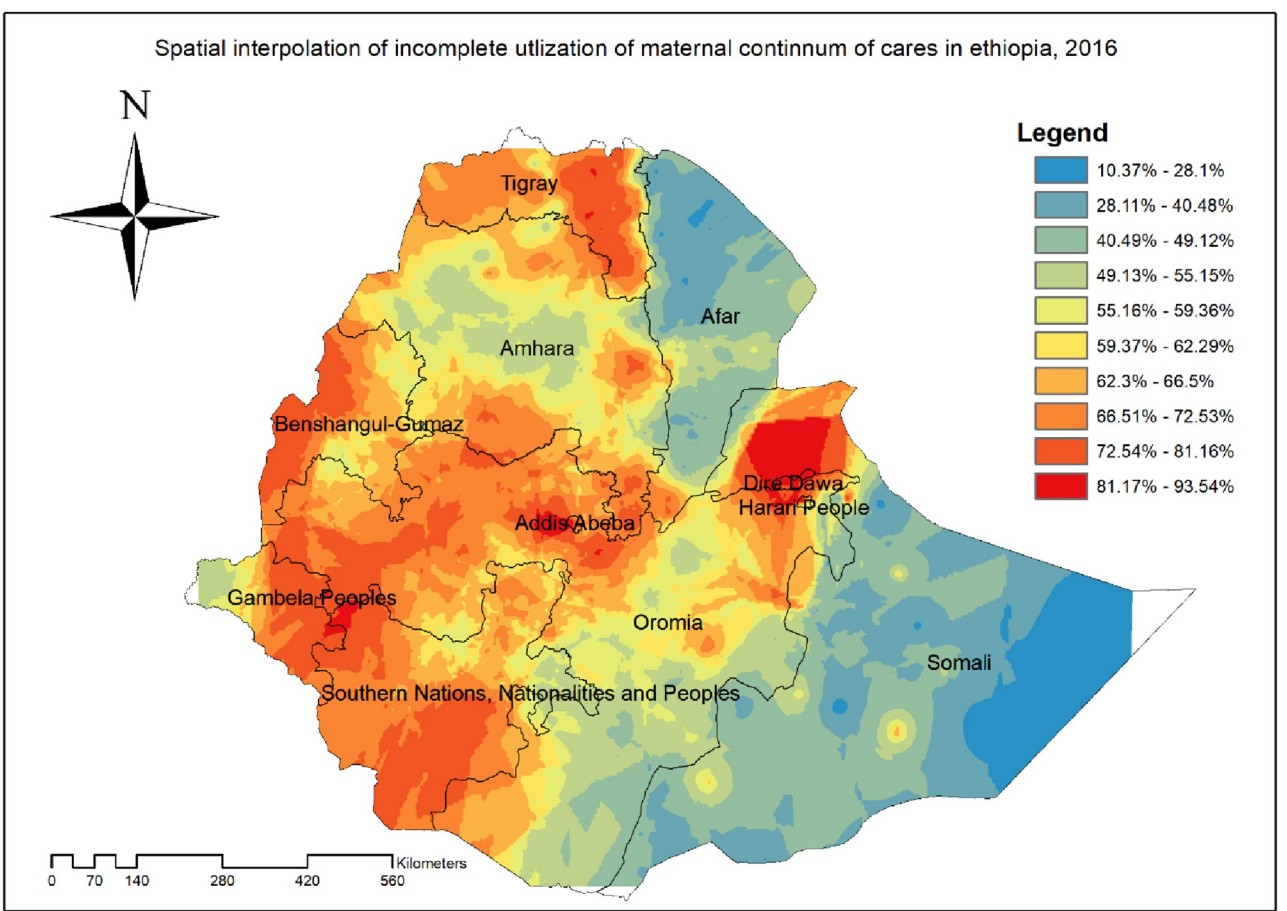

**Fig 5. Spatial scan statistical analysis of hotspot areas of incomplete utilization of maternal continuum of care in Ethiopia, produced using Arc GIS version 10.7.**

in the health facility which leads to the second delay for maternal mortality [48, 49]. It also imposes an extra cost for transportation as well as lack of availability of transportation and therefore they fail to attain the health facility to utilize maternal health service [50]. As a result, women who have barriers for accessing health cares may become less motivated to seek care compared with their counterparts.

Globally, evidence on the magnitudes and determinants of maternal continuum of care has grown substantially. This information has been used as a preventive measure that is linked to maternal and child health problems. From a policy point of view, the interventions which are designed to enhance to maternal health service utilizations such as dissemination information through mass media, women empowerment by providing education, improving their wealth status and creating opportunities to have their own jobs, and tackling barriers for accessing health service is needed to enable mothers for receiving the full continuum of maternal health services.

This study had both strength and limitations. The major strength of this study was it analyzed nationally representative data using large sample size and appropriate statistical analysis that help to detect the true effect and get reliable estimates and standard errors. Due to this, it could help to develop public health intervention or taking appropriate interventions to increase the utilization of maternal continuum of care. Despite the strengths, the findings of

**Table 4. Associated factors of incomplete utilization of maternal continuum of care among reproductive aged women in Ethiopia, 2016.**

| Variable | Category | COR (95% CI) | AOR (95% CI) |
|---|---|---|---|
| Residence | Urban | 1 | 1 |
| | Rural | 3.95 (2.89, 5.40) | 1.67 (1.02, 2.74)* |
| Educational status | No formal education | 1 | 1 |
| | Primary | 0.58 (0.43, 0.78) | 0.72 (0.52, 1.01) |
| | secondary | 0.31 (0.22, 0.44) | 0.57 (0.37, 0.86)* |
| | Higher | 0.20 (0.14, 0.29) | 0.48 (0.30, 0.77)* |
| Religion | Orthodox | 1 | 1 |
| | Protestant | 2.52 (1.63, 3.89) | 1.93 (1.25, 2.99)* |
| | Muslim | 2.19 (1.61, 2.99) | 1.46 (1.06, 2.01)* |
| | Other | 5.28 (0.85, 32.98) | 4.30 (0.66, 27.99) |
| Wealth index | poorest | 1 | 1 |
| | poorer | 0.48 (0.27, .86) | 0.52 (0.29, 0.95)* |
| | middle | 0.53 (0.29, 0.98) | 0.61 (0.32, 1.13) |
| | Richer | 0.30 (0.17, 0.53) | 0.39 (0.22, 0.72)* |
| | Richest | 0.15 (0.09, 0.25) | 0.47 (0.24, 0.92)* |
| Contraceptive use | Non-user | 1 | 1 |
| | Traditional user | 0.27 (0.12, 0.58) | 0.58 (0.27, 1.26) |
| | Modern user | 0.71 (0.56, 0.91) | 1.01 (0.78, 1.32) |
| Health insurance use | No | 1 | 1 |
| | Yes | 0.50 (0.32, 0.79) | 0.61 (0.40, 1.03) |
| Marital status | Single | 1 | 1 |
| | Married | 2.05 (1.15, 3.68) | 1.56 (0.86, 2.83) |
| | Widowed | 2.08 (0.64, 6.73) | 1.53 (0.46, 5.08) |
| | Divorced | 2.09 (0.93, 4.68) | 1.68 (0.74, 3.81) |
| Currently working | No | 1 | 1 |
| | Yes | 0.57 (0.45, 0.72) | 0.72 (0.56, 0.93)* |
| Parity | <5 | 1 | 1 |
| | > = 5 | 1.62 (1.22, 2.15) | 0.99 (0.70, 1.39) |
| Barriers for accessing health care | no | 1 | 1 |
| | Yes | 1.98 (1.55, 2.53) | 1.27 (1.12, 1.67)* |
| Mass media exposure | No | 1 | 1 |
| | Yes | 0.38 (0.29, 0.49) | 0.75 (0.55, 0.83)* |

this study need to be viewed in light of the following limitations. Even though relationships have been established, the findings cannot provide information on temporal relationships among the variables that are found to be associated with incomplete utilization of maternal continuum of care due to the cross-sectional nature of the EDHS data. This study might also have affected by social desirability bias and recall bias because the measurement for the main components of maternal continuum of care is self-reported based on women's recall response. Moreover, the study used circular window in the SaTScan analysis which is not able to detect irregular shaped clusters.

## Conclusion

In Ethiopia, incomplete utilization of maternal continuum of care had significant spatial variations across the country. Rural residency, Muslim and Protestant religious follower's, and presence of barriers for accessing health care were positively associated with incomplete utilization

of maternal continuum of cares. However, having better wealth index, education, mass media exposure, and being currently working were negatively associated with incomplete utilization of maternal continuum of care. Therefore, public health interventions targeted to enhance maternal service utilization and women empowerment by increasing maternal education awareness on maternal health services, creating job opportunities in hotspot areas of incomplete utilization of maternal continuum of care are crucial for reducing maternal morbidity and mortality in the country.

## Acknowledgments

We are grateful to thank the MEASURE DHS program for permitting us to obtain and use the data set for analysis.

## Author Contributions

**Conceptualization:** Tesfa Sewunet Alamneh, Achamyeleh Birhanu Teshale, Yigizie Yeshaw, Adugnaw Zeleke Alem, Hiwotie Getaneh Ayalew, Alemneh Mekuriaw Liyew, Zemenu Tadesse Tessema, Getayeneh Antehunegn Tesema, Misganaw Gebrie Worku.

**Data curation:** Tesfa Sewunet Alamneh, Achamyeleh Birhanu Teshale, Yigizie Yeshaw, Adugnaw Zeleke Alem, Hiwotie Getaneh Ayalew, Alemneh Mekuriaw Liyew, Zemenu Tadesse Tessema, Getayeneh Antehunegn Tesema, Misganaw Gebrie Worku.

**Formal analysis:** Tesfa Sewunet Alamneh, Achamyeleh Birhanu Teshale, Yigizie Yeshaw, Adugnaw Zeleke Alem, Hiwotie Getaneh Ayalew, Alemneh Mekuriaw Liyew, Zemenu Tadesse Tessema, Getayeneh Antehunegn Tesema, Misganaw Gebrie Worku.

**Methodology:** Tesfa Sewunet Alamneh, Achamyeleh Birhanu Teshale, Yigizie Yeshaw, Adugnaw Zeleke Alem, Hiwotie Getaneh Ayalew, Alemneh Mekuriaw Liyew, Zemenu Tadesse Tessema, Getayeneh Antehunegn Tesema, Misganaw Gebrie Worku.

**Software:** Tesfa Sewunet Alamneh.

**Supervision:** Yigizie Yeshaw.

**Writing – original draft:** Tesfa Sewunet Alamneh, Achamyeleh Birhanu Teshale, Yigizie Yeshaw, Adugnaw Zeleke Alem, Hiwotie Getaneh Ayalew, Alemneh Mekuriaw Liyew, Zemenu Tadesse Tessema, Getayeneh Antehunegn Tesema, Misganaw Gebrie Worku.

**Writing – review & editing:** Tesfa Sewunet Alamneh, Achamyeleh Birhanu Teshale, Yigizie Yeshaw, Adugnaw Zeleke Alem, Hiwotie Getaneh Ayalew, Alemneh Mekuriaw Liyew, Zemenu Tadesse Tessema, Getayeneh Antehunegn Tesema, Misganaw Gebrie Worku.

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
