## [Decision Letter · Decision Letter 0]

11 Jan 2022

PONE-D-21-39918Barriers for health care access affects maternal continuum of care utilization in Ethiopia; spatial analysis and generalized estimating equationPLOS ONE

Dear Dr. Alamneh,

Thank you for submitting your manuscript to PLOS ONE. After careful consideration, we feel that it has merit but does not fully meet PLOS ONE’s publication criteria as it currently stands. Therefore, we invite you to submit a revised version of the manuscript that addresses the points raised during the review process.

We look forward to receiving your revised manuscript.

Kind regards,

Sanjay Kumar Singh Patel, Ph.D.

Academic Editor

PLOS ONE

Journal Requirements:

3. We note that Figures 2 to 5 in your submission contain [map/satellite] images which may be copyrighted. All PLOS content is published under the Creative Commons Attribution License (CC BY 4.0), which means that the manuscript, images, and Supporting Information files will be freely available online, and any third party is permitted to access, download, copy, distribute, and use these materials in any way, even commercially, with proper attribution. For these reasons, we cannot publish previously copyrighted maps or satellite images created using proprietary data, such as Google software (Google Maps, Street View, and Earth). For more information, see our copyright guidelines: http://journals.plos.org/plosone/s/licenses-and-copyright.

a. You may seek permission from the original copyright holder of Figures 2 to 5 to publish the content specifically under the CC BY 4.0 license.  

Reviewers' comments:

Reviewer's Responses to Questions

**Comments to the Author**

1. Is the manuscript technically sound, and do the data support the conclusions?

Reviewer #1: Yes

Reviewer #2: Yes

2. Has the statistical analysis been performed appropriately and rigorously? 

Reviewer #1: Yes

Reviewer #2: Yes

3. Have the authors made all data underlying the findings in their manuscript fully available?

Reviewer #1: Yes

Reviewer #2: Yes

4. Is the manuscript presented in an intelligible fashion and written in standard English?

Reviewer #1: Yes

Reviewer #2: Yes

5. Review Comments to the Author

Reviewer #1: The manuscript entitled “Barriers for health care access affects maternal continuum of care utilization in Ethiopia; spatial analysis and generalized estimating equation” is an interesting study. Authors have assessed the spatial distribution of incomplete utilization of maternal continuum of care and its associated factors in Ethiopia based on 2016 Demographic and Health Survey data of Ethiopia. A total weighted sample of 4,772 reproductive age women were included. The study used ArcGIS and SaTScan software to explore the spatial distribution of incomplete utilization of maternal continuum of care. This study is noteworthy but the manuscript requires minor revision before its publication.

Comments

1, All Figures quality may be improved (high resolution).

2, At least one additional Figure (illustration) may be provided as to highlight the summary or prospect of this study.

3, Authors should discuss about the limitation to their study.

4, The English of manuscript can be polished (minor).

Reviewer #2: In this paper entitled "Barriers for health care access affects maternal continuum of care utilization in Ethiopia; spatial analysis and generalized estimating equation," the authors investigate the spatial distribution of incomplete utilization of maternal continuum of care and its associated factors in Ethiopia.". A total of 4772 reproductive-age women participated in the study. Using ArcGIS and SaTScan software, the usage of incomplete maternal continuum distribution is explored. The study is well carried out and is statistically sound. Moreover, the manuscript is easy to understand. However, This manuscript requires minor revision before its publication in PLOS One as follows:

Minor Comments:

1) Add the strength and limitations of the study in the manuscript.

2) The data is well presented in the manuscript and has undergone statistical rigor. But the author may improve the resolution of figures 1-5. The figures lack the clarity to make any conclusion. (Major)

3) The highlights the challenges and prospects of the study in the manuscript.

---

## [Author Response · Author response to Decision Letter 0]

11 Mar 2022

August 20121

Rebuttal letter

Manuscript ID: PONE-D-21-39918

Title: Barriers for health care access affects maternal continuum of care utilization in Ethiopia; spatial analysis and generalized estimating equation

Tesfa Sewunet Alamneh*, Achamyeleh Birhanu Teshale, Yigizie Yeshaw, Adugnaw Zeleke Alem, Hiwotie Getaneh Ayalew, Alemneh Mekuriaw Liyew, Zemenu Tadesse Tessema, Getayeneh Antehunegn Tesema, Misganaw Gebrie Worku

PLOS ONE

Dear Editor and reviewers, 

We would like to thank for your consideration and suggestion for the betterment our manuscript and make it more informative. We tried to amend the manuscript and address the questions raised by reviewer on it. Our point-by-point responses for each comment and questions are described in detail on the following pages. Further, the details of changes were shown by track changes in the supplementary document attached. 

Editor’s comment

Author’s response: Thank you dear editors for your concern. We tried to amend the format of the manuscript according to the journal guidelines

Author’s response: Thank you dear editors for your concern. We have putted this statement on data availability part. All the relevant data were included in the manuscript. However, it is ethically not acceptable to share the DHS data set to third parties and anyone who want the data set can access from the Measure DHS program at www.dhsprogram.com , through legal requesting. The authors had no special access privileges others would not have.

3. We note that Figures 2 to 5 in your submission contain [map/satellite] images which may be copyrighted. All PLOS content is published under the Creative Commons Attribution License (CC BY 4.0), which means that the manuscript, images, and Supporting Information files will be freely available online, and any third party is permitted to access, download, copy, distribute, and use these materials in any way, even commercially, with proper attribution. For these reasons, we cannot publish previously copyrighted maps or satellite images created using proprietary data, such as Google software (Google Maps, Street View, and Earth). For more information, see our copyright guidelines: http://journals.plos.org/plosone/s/licenses-and-copyright.

Author’s response: Thank you dear editors for your concern. The map is not copyrighted rather we have done using GIS software based on the shape files of Ethiopia received from Ethiopian Central Statistical Agency (CSA) by explaining the purpose of the study (https://africaopendata.org/dataset/ethiopia-shapefiles) and GPS data (longitude and latitude) from measure DHS program by explaining the objective of the study through online requesting and allow us to access the shape file and GPS data. Now we cite the source of the shape file since it is needed to explore the spatial distribution of incomplete utilizations of maternal continuum of care.

Author’s response: Thank you dear editors for your concern. We tried to change some references but we didn’t change reference from 23 up to 25 because they are books about spatial analysis. 

To reviewer 1

1. All Figures quality may be improved (high resolution) 

Author’s response: Thank you dear reviewer for your comment. As per your recommendation we tried to improve the quality of figures.

2. At least one additional Figure (illustration) may be provided as to highlight the summary or prospect of this study.

Author’s response: Thank you dear reviewer for your comment. We have illustrated it on the manuscript.

3. Comment: Authors should discuss about the limitation to their study.

Author’s response: Thank you dear reviewer, we tried to discuss the limitation of our study with in the study.

4. Comment: The English of manuscript can be polished (minor)

Author’s response: Thank you dear reviewer for your comment. We have checked the overall manuscript regarding language usage and grammar errors as and we try to amend it. In addition, we also consult language experts in our university and amendments were done based on their comments.

To reviewer 1

1. Add the strength and limitations of the study in the manuscript.

Author’s response: Thank you dear reviewer for your comment. We have included it in the last paragraph of discussion.

2. The data is well presented in the manuscript and has undergone statistical rigor. But the author may improve the resolution of figures 1-5. The figures lack the clarity to make any conclusion. (Major)

Author’s response: Thank you dear reviewer for your comment. As per your comment we tried to improve the resolution of all figures.

3. The highlights the challenges and prospects of the study in the manuscript.

Author’s response: Thank you dear reviewer for your comment. We have tried to address it by putting the following statements; Globally, evidence on the magnitudes and determinants of maternal continuum of care has grown substantially. This information has been used as a preventive measure that is linked to maternal and child health problems. From a policy point of view, the interventions which are designed to enhance to maternal health service utilizations such as dissemination information through mass media, women empowerment by providing education, improving their wealth status and creating opportunities to have their own jobs, and tackling barriers for accessing health service is needed to enable mothers for receiving the full continuum of maternal health services.

---

## [Editor Report · Decision Letter 1]

22 Mar 2022

Barriers for health care access affects maternal continuum of care utilization in Ethiopia; spatial analysis and generalized estimating equation

PONE-D-21-39918R1

Dear Dr. Alamneh,

We’re pleased to inform you that your manuscript has been judged scientifically suitable for publication and will be formally accepted for publication once it meets all outstanding technical requirements.

Kind regards,

Sanjay Kumar Singh Patel, Ph.D.

Academic Editor

PLOS ONE

---

## [Editor Report · Acceptance letter]

13 Apr 2022

PONE-D-21-39918R1 

Barriers for health care access affects maternal continuum of care utilization in Ethiopia; spatial analysis and generalized estimating equation 

Dear Dr. Alamneh:

I'm pleased to inform you that your manuscript has been deemed suitable for publication in PLOS ONE. Congratulations! Your manuscript is now with our production department. 

Kind regards, 

on behalf of

Dr. Sanjay Kumar Singh Patel 

Academic Editor

PLOS ONE